# Role of the Mre11 Complex in Preserving Genome Integrity

**DOI:** 10.3390/genes9120589

**Published:** 2018-11-29

**Authors:** Julyun Oh, Lorraine S. Symington

**Affiliations:** 1Biological Sciences Program, Columbia University, New York, NY 10027, USA; jo2410@columbia.edu; 2Department of Microbiology & Immunology, Columbia University Irving Medical Center, New York, NY 10032, USA

**Keywords:** Mre11, Rad50, Xrs2/Nbs1, Sae2/Ctp1/CtIP, Tel1/ATM, MRX/N, DSB, DNA damage checkpoint, homologous recombination, DNA repair

## Abstract

DNA double-strand breaks (DSBs) are hazardous lesions that threaten genome integrity and cell survival. The DNA damage response (DDR) safeguards the genome by sensing DSBs, halting cell cycle progression and promoting repair through either non-homologous end joining (NHEJ) or homologous recombination (HR). The Mre11-Rad50-Xrs2/Nbs1 (MRX/N) complex is central to the DDR through its structural, enzymatic, and signaling roles. The complex tethers DNA ends, activates the Tel1/ATM kinase, resolves protein-bound or hairpin-capped DNA ends, and maintains telomere homeostasis. In addition to its role at DSBs, MRX/N associates with unperturbed replication forks, as well as stalled replication forks, to ensure complete DNA synthesis and to prevent chromosome rearrangements. Here, we summarize the significant progress made in characterizing the MRX/N complex and its various activities in chromosome metabolism.

## 1. Introduction

Genome integrity is constantly threatened by exogenous and endogenous stresses that can result in various types of DNA damage. Double-strand breaks (DSBs), which can arise spontaneously when a replication fork collapses or can be induced by exposure to genotoxic agents such as ionizing radiation (IR), are one of the most cytotoxic forms of DNA damage. Failure to repair a DSB results in loss of genetic information or even cell death, whereas inaccurate repair can generate chromosome rearrangements, such as translocations, inversions, or copy number variations. Even though accidental DSBs pose a significant threat to genome stability, DSBs are necessary intermediates in a number of programmed recombination events, including V(D)J recombination during lymphocyte development, meiosis, and mating-type switching in budding yeast. In all cases, DSBs need to be properly detected and repaired to preserve genomic integrity.

Eukaryotic cells have evolved a sophisticated and highly conserved DNA damage response (DDR) system, which consists of a kinase cascade in response to lesion recognition coordinated with various repair mechanisms, to cope with DSBs. The Mre11-Rad50-Xrs2/Nbs1 complex (MRX in budding yeast, MRN in organisms with Nbs1 replacing Xrs2) orchestrates all stages of the DDR, including sensing the initial lesion, activating checkpoint signaling, driving specific repair pathways, and structurally bridging the participating DNA molecules together. The *MRE11*, *RAD50*, and *XRS2* genes were originally identified by their requirement for IR resistance and meiotic recombination in *Saccharomyces cerevisiae* (budding yeast) [1]. Mre11 and Rad50 are conserved in all domains of life whereas Xrs2/Nbs1 is less conserved than Mre11 and Rad50 and has only been identified in eukaryotes [2]. The proteins form a heterohexameric DNA binding complex containing dimers of each subunit [3,4]. Germline hypomorphic mutations of human MRN complex components are associated with Nijmegen breakage syndrome (NBS), NBS-like disorder and ataxia telangiectasia-like disorder (ATLD), which are characterized by cellular radiosensitivity, immune deficiency, and cancer proneness [2,5,6,7]. In mammals, the MRN complex is essential for cell viability, unlike in yeast, in which the null mutations are viable [2]. In this review, we focus on studies performed in *S. cerevisiae* and refer to studies in other systems where appropriate.

## 2. Various Roles of the MRX/N Complex in DNA Damage Recognition and Repair

### 2.1. Double-Strand Break Detection and Checkpoint Activation

The cellular response to DSBs is initiated when the MRX/N complex binds to the broken DNA ends within minutes of their generation [8]. MRX/N is normally diffused evenly throughout the nucleus until a lesion induces the redistribution of the proteins to the damaged site in high concentration [9,10,11], indicating that the complex normally surveys the nucleus for a binding site. MRX/N scans along the DNA via facilitated diffusion to detect free ends [12]. After binding to DSBs, MRX/N recruits the transducing kinase Tel1/ATM and activates the DNA damage checkpoint-signaling cascade (Figure 1) [2]. Tel1/ATM is a member of the PIKK family, characterized as a serine/threonine protein kinase with an N-terminal HEAT repeat domain and C-terminal kinase domain [13]. Mutations in the ATM gene are associated with ataxia telangiectasia (A-T), a human syndrome characterized by neurodegeneration, sensitivity to IR, immunodeficiency, and predisposition to cancer [14]. The cellular phenotype of A-T is similar to NBS and ATLD even though the clinical manifestations of the diseases differ. The N-terminal HEAT domain of Tel1/ATM physically interacts with the conserved Tel1/ATM-interacting motifs at the C-terminus of Xrs2/Nbs1 [15,16,17,18]. Tel1/ATM kinase activity is stimulated by MRX/N binding to protein-bound DNA ends but is independent of the Mre11 nuclease activity [19,20]. ATP-driven conformational changes in the MRX/N complex have been shown to regulate Tel1/ATM kinase activity [21,22,23], suggesting that distinct allosteric effects mediate Tel1/ATM activation. Human ATM undergoes intermolecular autophosphorylation upon interaction with the MRN complex, which results in dissociation of the inactive dimer into active monomers [24]. The exact molecular mechanism of ATM/Tel1 activation remains to be elucidated.

With appropriate cell cycle cues, the MRX/N complex initiates degradation of 5′-terminated strands at DSBs resulting in formation of long tracts of single-stranded DNA (ssDNA), a process termed end resection. End resection follows a two-step, bidirectional mechanism (Figure 1). First, MRX/N, stimulated by its cofactor Sae2/CtIP (CtIP is the functional ortholog of Sae2 in mammalian cells), catalyzes an endonucleolytic cleavage internal to 5′ ends, generating a single-stranded nick [25,26]. The nick is an entry site for two parallel extensive resection machineries, the Exo1 5′-3′ exonuclease or the Sgs1/BLM/WRN helicase together with the Dna2 endonuclease, which will further degrade in a 5′ to 3′ direction [27,28,29,30,31]. Meanwhile, MRX/N utilizes its 3′ to 5′ exonuclease activity to proceed back toward the DSB ends [30,32,33]. The resulting ssDNA overhangs are bound by replication protein (RPA). The Mec1/ATR kinase, another PIKK family member, is recruited to RPA-bound ssDNA via Ddc2/ATRIP to continue checkpoint signaling [34]. Several proteins, including the 9-1-1 damage clamp, Dpb11/TOPBP1, Dna2 (budding yeast), and ETAA1 (human cells), contribute to Mec1/ATR activation [35,36]. The two master DDR kinases, Tel1/ATM and Mec1/ATR, phosphorylate histone H2A (H2AX in mammals) in the vicinity of DSBs, creating a platform for recruitment of DNA repair factors and modulating chromatin accessibility [37,38]. The checkpoint mediator Rad9/53BP1 binds to phosphorylated H2A/H2AX (γH2A/H2AX) and other chromatin marks, and is itself a substrate for Tel1/ATM and Mec1/ATR-dependent phosphorylation. Phosphorylated Rad9 recruits the downstream checkpoint effector kinase, Rad53/CHK2, to damage sites where it can be activated by Tel1 or Mec1-dependent phosphorylation [35]. The signaling cascade arrests cell-cycle progression to ensure enough time for repair to be completed, regulates activities of repair factors and induces a transcriptional response to DNA damage [35].

The attenuation of Tel1/ATM kinase activity temporally correlates with the initiation of end resection, which in turn activates Mec1/ATR [39,40]. Thus, MRX plays a critical role in the DDR as a sensor for DSBs, and in the transition from Tel1/ATM to Mec1/ATR signaling. In budding yeast, Tel1 checkpoint activity is functionally redundant with Mec1 activity but Mec1 is considered to be the principal PIKK. Lack of Tel1 does not sensitize cells to DNA damaging agents and only slightly reduces end resection efficiency [41,42]. Moreover, damage-induced phosphorylation of Rad53 is unperturbed in the *tel1Δ* mutant as long as Mec1-dependent checkpoint pathway is active, unlike in mammals where ATM is required for CHK2 activation [41]. This minor role of Tel1 in DSB signaling may reflect the efficient initiation of resection in *S. cerevisiae* and, thus, the preference to stimulate Mec1 kinase activity. Indeed, if end resection is delayed and MRX persists at ends, then Te1l becomes hyper-activated [43,44]. Tel1-dependent phosphorylation of Mre11 and Xrs2 in response to DNA damage has been observed [44,45], though its functional significance is not completely understood. Elimination of all the SQ and TQ motifs (preferred targets for PIKKs) in Xrs2 does not cause any defects in telomere maintenance or the DDR [46]. In mammalian cells, ATM-dependent phosphorylation of Nbs1 appears to be necessary for S phase checkpoint signaling [47,48]. ATM and ATR dependent phosphorylation of *Xenopus* Mre11 triggers the dissociation of MRN from chromatin, suggesting that it acts as a negative feedback loop to limit MRN activity [49].

### 2.2. Role of MRX/N in Double-Strand Break Repair

There are two mechanistically distinct pathways to repair DSBs: non-homologous end joining (NHEJ) and homologous recombination (HR) (Figure 2). HR utilizes an intact donor template with extensive sequence homology for mostly accurate repair. In contrast, NHEJ directly re-ligates the two ends of the broken DNA, which can be accompanied by gain or loss of nucleotides at the junction. Both mechanisms are highly conserved throughout eukaryotic evolution.

#### 2.2.1. Non-homologous end joining

Repair by NHEJ initiates with Ku heterodimer (Ku70-Ku80) binding to the DSB ends to prevent exonucleolytic degradation by Mre11 and Exo1 [32,33,50,51]. Ku recruits the DNA ligase IV complex (Lig4/Dnl4-Lif1/Xrcc4-Nej1/Xlf), which then directly ligates the two broken ends [52]. End joining can be precise or imprecise, depending on how the ends are processed before ligation [52]. Vertebrate cells require additional factors to facilitate NHEJ, including the catalytic subunit of DNA-dependent protein kinase (DNA-PKcs), another member of the PIKK family, and Artemis [53]. MRX is essential for NHEJ in budding yeast [54,55]. In vitro, MRX tethers DNA ends and stimulates their ligation by the DNA ligase IV complex [56]. Moreover, the Xrs2 subunit of the complex directly interacts with Lif1 in vitro, suggesting that it collaborates with Ku to recruit the DNA ligase IV complex to DSBs [56,57,58]. In mammalian cells, MRN plays a supporting role in classical and alternative NHEJ (alt-NHEJ) [59,60,61,62].

In the absence of classical NHEJ factors, a distinctive alt-NHEJ pathway can occur using microhomologies (MHs) (5–25 nt in yeast; 1–16 nt in mammalian cells) flanking the DSB [63,64,65,66,67,68]. MH-mediated end joining (MMEJ) requires end resection to expose the MHs internal to the break ends, the MHs guide annealing, and after flap trimming and gap filling the ends are sealed by DNA ligase I or III [68] (Figure 2). While resection initiation by MRN and CtIP is essential for MMEJ in mammalian cells, MMEJ is still detected in yeast cells lacking MRX and Sae2, presumably due to the use of other resection mechanisms (see below) [59,66,69,70,71,72]. The single-strand annealing (SSA) mechanism is mechanistically similar to MMEJ except the repeats flanking the DSB tend to be longer and annealing of exposed complementary single-stranded regions is catalyzed by Rad52. The SSA pathway can be very efficient in budding yeast, particularly when it involves long (>200 bp) direct repeats and strand invasion is prevented by a *rad51Δ* mutation. Cells lacking MRX exhibit a delay in product formation and reduced efficiency of SSA [28,73].

#### 2.2.2. End Resection and Homologous Recombination

End resection is essential to generate ssDNA for Rad51 binding and subsequent steps of HR, as well as removing the preferred substrate for NHEJ [74,75]. Thus, end resection is a key step in determining which pathway is used to repair DSBs. End resection is regulated by cyclin-dependent kinases (CDK) during the cell cycle to ensure that commitment to HR occurs when a sister chromatid is available for homology-directed repair [76,77]. The initial endonucleolytic cleavage by MRX/N removes Ku from ends and creates an ssDNA overhang that is sufficient for HR [12,32,33,78]. MRX/N-catalyzed end clipping is also critical for removing hairpin-capped ends and covalently-bound proteins from DSB ends [79,80]. Besides its catalytic function in resection, the MRX complex physically recruits Exo1, Dna2, and Sgs1 to the damaged site [81]. In budding yeast, mutations that eliminate the Mre11 nuclease activity (*mre11-nd*) or *sae2Δ* confer a mild resection defect at DSB ends with no covalent modification because, in the absence of MRX-catalyzed endonucleolytic cleavage, Exo1 and Sgs1-Dna2 can directly degrade DSB ends, albeit with some delay [82]. However, *mre11Δ, rad50Δ*, and *xrs2Δ* mutants show a more severe resection defect, presumably because the absence of the complex attenuates recruitment of Sgs1 and Dna2 to DSBs [81]. The DNA damage sensitivity and end resection defects of *mre11Δ*, *mre11-nd*, and *sae2Δ* cells can be rescued by deletion of *YKU70* or *YKU80* (encoding the Ku complex) in an *EXO1* dependent manner, indicating that Ku prevents access to DNA ends by Exo1 [51,81,83]. Furthermore, *exo1Δ* synergizes with *mre11Δ* for end resection defects and IR sensitivity [84,85,86]. Although recruitment of Exo1 to ends is reduced in *mre11Δ* and *rad50Δ* mutants [81], Exo1 is clearly able to act in the absence of MRX, especially when Ku is eliminated from cells. By contrast, *mre11Δ* and *sgs1Δ* mutations are epistatic for IR sensitivity, consistent with an important role for MRX in recruiting Sgs1-Dna2 to DSBs [51]. In the absence of Mre11 nuclease, resection of endonuclease-induced DSBs is mainly due to Sgs1-Dna2. Thus, the *mre11-H125N sgs1Δ* double mutant exhibits greatly increased DNA damage sensitivity and delayed resection relative to the single mutants [51,81,87]. MRX/N directly stimulates the nuclease activities of Sgs1-Dna2 and Exo1 in vitro suggesting cooperation between the short range and long-range resection machineries [32,81,88,89,90]. In *Schizosaccharomyces pombe* (fission yeast) and mammalian cells, *mre11-nd* mutations confer greater defects in resection, HR, and resistance to DNA damaging agents than is observed in budding yeast [20,78,91]. This difference may be due to differing ability of Sgs1-Dna2 to compensate for MRN [78,92], or Ku may be more dominating in fission yeast and higher eukaryotes, thereby posing a greater barrier for Exo1.

In *S. cerevisiae*, HR still occurs in *mre11Δ*, *rad50Δ*, and *xrs2Δ* mutants. By contrast, *rad51Δ* and *rad52Δ* mutants are completely deficient for most types of HR even though they exhibit comparable radiation sensitivity to the *mre11Δ* mutant. The lack of MRX delays but does not prevent mating-type switching, a specialized mitotic intrachromosomal gene conversion process [93]. Similarly, DNA damage-induced sister-chromatid recombination is reduced, but not eliminated in cells lacking MRX [94,95]. In diploid yeast cells, the absence of MRX increases the rate of spontaneous heteroallelic recombination about 10-fold [94,96,97,98]. In the *S. pombe rad50* mutant, spontaneous recombination between direct repeats is decreased by ~15-fold while the frequency of interchromosomal recombination in diploid cells is greatly increased [99]. The inter-chromosomal hyper-recombination phenotype is suggested to result from channeling of repair from sister chromatids to homologs. Together, these observations suggest that there can be sufficient resection of DNA ends lacking covalent modifications in MRX-deficient cells to mediate Rad51 nucleoprotein filament formation and strand invasion. Elimination of all resection mechanisms (*mre11Δ exo1Δ sgs1Δ* or *mre11-nd exo1Δ sgs1Δ* triple mutants) prevents HR and causes cell lethality in budding yeast [28]. The long-range resection mechanisms are not essential for HR, but ensure fidelity by preventing recombination between short dispersed repeats, and are important to generate ssDNA for checkpoint signaling via Mec1/ATR [27,29,100].

### 2.3. Meiotic Recombination

During meiosis, formation and repair of programmed DSBs ensures correct alignment and segregation of chromosome homologs in addition to generating diversity [101]. In budding yeast, the MRX complex plays at least two roles during meiotic recombination. First, the complex is required for the meiosis-specific topoisomerase-like protein, Spo11, to generate DSBs. After DSB formation, Spo11 is covalently attached to the 5′ ends at break sites. The second role of the Mre11 complex is to remove Spo11 from break ends by endonucleolytic cleavage, releasing Spo11 attached to a short oligonucleotide [80]. *mre11-nd*, *sae2Δ*, and *rad50S* mutants are proficient for meiotic DSB formation, but are defective in removing Spo11 from ends (reviewed in [82]), while lack of the complex prevents Spo11-mediated meiotic DSB formation. By coupling DSB formation and processing, MRX ensures timely and efficient crossover formation and restoration of genome integrity before the meiotic division. Unlike in budding yeast, fission yeast MRN is not required for meiotic DSB formation but is strictly required for DSB processing [102,103]. Interestingly, Mre11 and Rad50 are required for both meiotic DSB formation and processing in *Caenorhabditis elegans*, whereas Nbs1 is only required for the latter [104]. The evolutionary reason for this differential requirement for MRX/N in meiotic DSB formation in different organisms is currently unknown.

Removal of Spo11 by MRX creates a substrate for Exo1-mediated degradation. In contrast to mitotic DSB processing, the Sgs1-Dna2 pathway does not appear to play an active role in meiotic end resection [105,106]. The bidirectional model of end resection was proposed based on the phenotype of *mre11* and *exo1* mutants during meiosis. Diploid cells expressing an *mre11* allele that encodes a protein proficient for endonuclease but defective for exonuclease activity in vitro (*mre11-H59S*) generate longer (41–300 nt) Spo11-linked oligonucleotides compared to wild-type (WT) cells (12–40 nt) [30]. In mutants lacking Exo1 nuclease activity (*exo1-nd*), average resection tract length is ~300 nt, compared with ~800 nt in WT cells [105,107], longer than predicted if MRX-Sae2 clipping removed only 12–40 nt from DSB ends. The *mre11-H59S exo1Δ* double mutant accumulates DSBs and crossover formation is delayed [30]. These observations are consistent with bidirectional resection by MRX and Exo1 in which the Mre11 exonuclease is responsible for processing back towards the DSB ends from the single-stranded nick created by the Mre11 endonuclease ~300 nt internal to the Spo11-bound end while Exo1 proceeds in a 5′-3′ direction away from the DSB. Although resection tracts are only ~300 nt long in *exo1-nd* diploids, sporulation, and spore viability are unaffected indicating that MRX-catalyzed resection is sufficient for meiotic recombination [105].

### 2.4. Hairpin Resolution

SbcD and SbcC, the *Escherichia coli* homologs of Mre11 and Rad50, respectively, function primarily to cleave DNA hairpins formed by palindromic sequences [108]. DNA hairpins form on the lagging strand during DNA replication, and after SbcCD cleavage the resulting DSB is repaired by RecA-dependent sister-chromatid recombination, preserving the palindrome [109,110]. Consistently, hairpin-capped DNA ends are resolved by MRX-Sae2 to prevent palindromic gene amplification and other chromosomal rearrangements in yeast [79,111,112,113,114,115]. In vitro, the purified MRX complex has weak activity at hairpin-capped DNA ends. However, RPA bound to the unpaired ssDNA region of the hairpin triggers MRX and Sae2-dependent nicking, similar to the reaction observed at protein-blocked DSB ends [33].

### 2.5. Replisome Stability

DNA replication forks are fragile structures where damage can easily occur. Indeed, Rad52 foci form without any exogenous DNA damage in approximately 5% of cells during S phase [116,117], corresponding to spontaneous DNA damage during replication. The DNA damage and replication checkpoints are responsible for stabilizing the fork, resolving replicative stress, and completing DNA synthesis in order to avoid chromosomal rearrangements that lead to genomic instability [118].

Given its pivotal role in the DDR, it is not a surprise that MRX/N also plays a crucial role in responding to replication stress. In mammalian cells, the MRN complex colocalizes with proliferating cell nuclear antigen (PCNA) and with sites of BrdU incorporation throughout an unperturbed S phase [119]. Mre11 is also detected at replication-origin-proximal sites by chromatin immunoprecipitation (ChIP) and by isolation of proteins on nascent DNA (iPOND), and its enrichment significantly increases near stalled forks [119,120]. In *Xenopus laevis*, depleting the Mre11 complex during replication results in increased DNA breakage and DSB accumulation [121].

The Mre11 complex is most likely recruited to replication forks by RPA. Stable interaction between MRN and RPA has been observed at replication sites throughout S phase with increased interaction at sites of stalled forks induced by hydroxyurea (HU) or ultraviolet light (UV) [122,123]. A mutation in the N-terminal oligonucleotide/oligosaccharide-binding (OB) fold of the Rfa1 subunit of yeast RPA (*rfa1-t11*) abolishes MRX recruitment to replication forks, leading to fork collapse and separation of sister chromatids [124]. The MRX complex stabilizes the association of essential replisome components at stalled forks independently of the S phase checkpoint or the nuclease activity of Mre11 [125]. Disruption of the MRX complex results in defective fork recovery and failure to properly complete DNA replication under replication stress [125]. These observations indicate a structural contribution of the MRX complex during replication.

When a replication fork encounters a lesion, the fork may undergo reversal to form a four-way junction, which protects the fork and allows for restart [126]. Two SWI/SNF-family translocases, ZRANB3 and SMARCAL1, have been shown to mediate such fork remodeling in mammalian cells [127,128,129]. Fork reversal generates a one-ended DSB that is subjected to degradation by Mre11 and Exo1. Rad51 accumulation at stalled forks depends on Mre11 nuclease activity, suggesting a requirement for end resection [120]. Degradation by Mre11 at stalled forks may promote template switch-mediated repair or may amplify checkpoint signaling by enlarging ssDNA gaps. However, uncontrolled Mre11 nuclease activity can lead to extensive nascent strand degradation [130,131,132]. Mre11-mediated degradation of the reversed fork in *Brca*-deficient cells causes genome instability upon treatment with replication stress, demonstrating the toxicity of nucleolytic degradation at unprotected stalled forks [133]. Such catastrophe is normally prevented by BRCA1 and BRCA2 in mammals, which protect nascent ssDNA from Mre11-mediated degradation by promoting formation of stable Rad51 nucleoprotein filaments [127,128,132,134,135,136]. Similarly to DSB end resection, CtIP/Ctp1 initiates Mre11-dependent degradation of reversed forks and Exo1 further extends the resection tracts [135,137,138]. Tel1 has also been reported to regulate the nuclease activity of Mre11 at replication forks that reverse after topoisomerase poisoning [139].

### 2.6. Cohesin Loading and/or Stabilization by MRX/N

Cohesin is a structural maintenance of chromosomes (SMC) complex that maintains sister chromatid pairing until mitosis. In addition to ensuring proper chromosome segregation after replication, cohesin also contributes to DSB and stalled replication fork repair, presumably by maintaining sister chromatids in a conformation that favors efficient HR [140,141,142]. DDR factors, including MRX, Tel1, and Mec1, regulate cohesin recruitment to DSBs and stalled forks [141,143,144,145,146]. Mec1- and Tel1-dependent phosphorylation of histone H2A generates a large domain that spreads from DSBs and enables cohesin binding [145]. How MRX/N facilitates cohesin loading is unclear, as no physical interaction has been observed between the two complexes. Interestingly, structural features of Rad50 that are important for bridging sister chromatids, including the hook and coiled-coil domains, are important for facilitating cohesin loading to forks during replication stress [147]. MRX/N itself is structurally capable of and sufficient to hold sister chromatid together at DSBs at early time points [124], suggesting that cohesin may simply load on to sisters initially held together by MRX/N. Nonfunctional or insufficient cohesin at damaged site causes sensitivity to genotoxins, and leads to increased recombination between homologs instead of sister chromatids [142,143,145].

### 2.7. Prevention of Gross Chromosome Rearrangements

A hallmark of cancer is the accumulation of gross chromosome rearrangements (GCRs), such as translocations, interstitial or terminal deletions, gene amplifications and inverted duplications [148,149]. Genetic assays in *S. cerevisiae* that allow quantitative measurement of the accumulation of GCRs as well as identification of the spectrum of events have been useful for identifying pathways that normally suppress GCR formation [150]. The rate of GCRs increases by around 600-fold in the absence of *MRE11*, *RAD50*, or *XRS2* [151]. This effect is approximately 60-fold higher than that observed for simultaneous inactivation of *RAD51* and *YKU70,* suggesting that the MRX complex acts beyond facilitating HR and NHEJ to prevent GCRs [151]. *mre11-nd* and *sae2Δ* mutants show 13-fold and 5-fold increased GCR rates, respectively, [112] and loss of Tel1 has no significant effect on the GCR rate [152,153], indicating that the enzymatic and signaling functions of the MRX complex only mildly contribute to GCR suppression. These data suggest that the structural and/or replisome stabilization functions of MRX could be important to suppress GCRs. While most GCRs from normal cells are due to loss of terminal sequences and de novo telomere addition, GCRs recovered from *mre11Δ* cells are mostly due to MH-mediated translocations or inverted duplications [151]. The major class of GCR recovered from *mre11-nd* and *sae2Δ* mutants is hairpin-mediated inverted duplication, a rare event in WT cells [112,154]. Inverted duplications are thought to result from pairing between short (5–9 bp) natural inverted repeats exposed by end resection of a spontaneous DSB, fill-in DNA synthesis to generate a hairpin-capped chromosome, replication to form a dicentric chromosome, followed by breakage and homology-dependent repair using dispersed repeats. Mre11 endonuclease and Sae2 are suggested to prevent generation of inverted duplications by cleaving the hairpin-capped chromosome intermediate [112,113,114].

### 2.8. Telomere Maintenance

In addition to its role in the DDR, MRX/N is required for maintaining telomere length homeostasis. Telomeres are specialized nucleoprotein structures that cap the ends of each chromosome to prevent degradation and fusions [155]. Telomerase maintains the length of telomeres by adding telomeric repeats to the terminal ssDNA G-rich tail using an RNA template. In budding yeast, recruitment of telomerase to short telomeres requires MRX and Tel1. As in DSB repair, MRX first binds to telomeric ends and recruits Tel1, which in turn recruits telomerase through phosphorylation of Cdc13 [156,157]. Null mutations in any of the components of the MRX complex results in short telomeres similar to the phenotype of *tel1Δ* cells [158,159]. Tel1 association to telomeres is counteracted by Rif2, which negatively regulates telomere length [160,161]. Rif2 and Tel1 both interact with C-terminus of Xrs2 [162], suggesting that Rif2 may inhibit Tel1 localization to telomeres by interfering with MRX-Tel1 interaction. Unlike in *S. cerevisiae*, mammalian ATM is not required for association of telomerase to short telomeres [163]. However, MRN has been shown to mediate ATM-dependent response at dysfunctional telomeres [164]. In addition, MRX/N is essential for processing telomeric DNA ends to generate 3′ ssDNA overhangs, which impairs NHEJ and prevents fusions [165].

## 3. Structural and Biochemical Properties of the MRX Complex

### 3.1. Mre11 and Rad50

Mre11 interacts independently with both Rad50 and Xrs2/Nbs1, and dimerizes with itself to form the core of the complex (reviewed in [1]). Mre11 has phosphodiesterase motifs in the N-terminal region of the protein and displays manganese-dependent ssDNA endonuclease and 3′-5′ double-stranded DNA (dsDNA) exonuclease activities in vitro (Figure 3A) (reviewed in [166]). Alone, Mre11 endonuclease is only active on ssDNA, but in the context of MRX/N and phosphorylated Sae2/CtIP, it cleaves the 5′ terminated strands of linear dsDNA [25,26]. Mre11-dependent endonucleolytic cleavage is stimulated by proteins bound to DNA ends, such as Ku, or even by RPA bound to a short ssDNA overhang [12,32,33]. MRX-Sae2 cleaves 15–20 nt from the ends of linear dsDNA with biotin-streptavidin adducts at the 5′ ends, whereas in vivo studies indicate that incision occurs much further from DNA ends [25,30,105]. The positions of MRX and Sae2-dependent nicks are likely influenced by chromatin structure around DSBs [33,107]. Mutations in the conserved residues within the phosphodiesterase domain (e.g., Asp16, Asp56, His125, and His213 of ScMre11) completely eliminate endo- and exo-nuclease activities in vitro (reviewed in [1]). *mre11-nd* mutants with intact complex stability are proficient for telomere length maintenance and DDR, showing only mild IR sensitivity [167]. However, *mre11-nd* mutants exhibit severe defects in meiosis and hairpin resolution, similar to *mre11Δ* cells, highlighting the importance of the nuclease activity for these functions [79,168,169]. Mre11 forms a dimer in solution mediated via the phosphodiesterase domain [170]. Disrupting Mre11 dimerization sensitizes cells to DNA damaging agents, demonstrating the importance of the dimeric form in the overall complex function [91,171]. Moreover, structural analysis suggests that Mre11 dimerization coordinates short-range DNA bridging [91].

Rad50 is a member of the SMC family of proteins, characterized by ATPase motifs at the N and C termini separated by two long coiled-coil domains (Figure 3A) [2]. The coiled coils, which are separated by a zinc binding CxxC motif referred to as a zinc hook, fold back on themselves to form antiparallel intramolecular coiled coils juxtaposing the Walker A and B ATPase motifs to generate an ATP nucleotide binding domain. The zinc hook at the apex of the coiled-coil domains facilitates dimerization with a second hook domain via the chelation of a zinc ion (Figure 3) [172,173]. Additional contacts between the coiled-coils stabilize intramolecular dimerization of Rad50 [174]. The globular DNA binding domain of the Mre11-Rad50 complex is comprised of an Mre11 dimer associated with the ATPase cassettes of a Rad50 dimer [175,176,177]. ATPase activity triggers conformational changes in the Mre11-Rad50 complex, which are crucial for regulating the nuclease activity and the diverse functions of the complex (Figure 3B). ATP binding induces the ‘closed’ conformation, in which Rad50 head domains dimerize and block the nuclease active site of Mre11. In this conformation, DNA tethering, ligation, and Tel1/ATM activation are promoted. ATP hydrolysis drives a disengagement of the Rad50 dimer, allowing the Mre11 active site to access DNA for its nuclease activity, thus promoting end resection [178,179,180,181]. Consistently, the hairpin opening activity and endonucleolytic cleavage internal to a protein-blocked DNA ends require ATP in vitro [25,170,182]. Mutating the ATP-binding domain of Rad50 results in null-phenotype, highlighting the importance of conformational changes for MR complex functions [175,183]. Mutation of residues near the ATPase domain generated a class of *rad50* mutants, referred to as *rad50S*, which confer a similar phenotype to *mre11-nd* and *sae2Δ* mutants [183]. A recent study identified a physical interaction between Rad50 and phosphorylated Sae2, which is eliminated by the rad50S mutation, suggesting that the *rad50S* end-clipping defect is due to impaired regulation of MR by Sae2 [184].

The hook and coiled-coil-mediated dimerization of the Mre11 complex is suggested to tether the two DSB ends and to hold sister chromatids together (Figure 4) [95,124,185]. Maintaining close proximity of DNA ends may promote NHEJ by stimulating ligation. Indeed, the MRX complex directly stimulates the activity of the yeast DNA ligase IV complex in vitro, consistent with the need for end tethering to promote NHEJ [56]. Bridging sister chromatids at DSBs is expected to facilitate the homology search during HR and prevent the damaged chromatid from physically separating from the rest of the chromosome. Consistent with this view, the Rad50 hook domain is crucial to prevent a DSB from becoming a chromosome break [95,185,186]. While some studies have shown that reduced end tethering correlates with reduced efficiency of homology-dependent DSB repair [187,188], a recent study found no defect in gene conversion or SSA when end tethering is compromised by loss of the Xrs2 subunit [189].

The coiled-coils could extend up to approximately 500 Å, or 1000 Å in the hook-mediated dimeric state [190,191]. The length of the coiled coil is critical for the Mre11 complex functions. Truncation of the coiled-coil domain abolishes meiotic DSB formation and telomere length maintenance [95]. Interestingly, shortening the coiled-coil domains affects HR and NHEJ functions of the Mre11 complex differently. While truncation of ~300 amino acids severely impairs NHEJ, HR-related functions are largely intact [95]. Shortening the domain further (~500 amino acids removal) completely abolishes HR, suggesting that flexibility in the coiled-coils is important for mediating recombination [95]. As noted above, oligomerization between the coiled-coils could contribute to the end tethering function of MRX and explain the phenotype of the shortened coiled-coil mutants [174].

### 3.2. Xrs2/Nbs1

Xrs2/Nbs1 is the eukaryote-specific component of the Mre11 complex and harbors a number of protein–protein interaction motifs (Figure 3A), suggesting it functions as a scaffold to coordinate repair and signaling activities [5,16]. It is the only component of MRX harboring a nuclear localization signal (NLS) and its interaction with Mre11 is necessary for translocation of MR into the nucleus [5,16,192]. Consistently, NBS patients with the 657del5 allele, which results in reduced amounts of the truncated protein, have Mre11 mislocalized to the cytoplasm [5,193]. Mutation of residues within the Mre11 interaction motif that prevent Xrs2/Nbs1 binding to Mre11 confers a phenotype indistinguishable from *xrs2Δ* and *mre11Δ* null mutations in budding yeast [16], and leads to embryonic lethality in mice [194]. The requirement for Xrs2 to translocate Mre11 to the nucleus can by bypassed by expressing an Mre11-NLS fusion protein in budding yeast [16,195]. Nuclear Mre11 partially restores DNA damage resistance to *xrs2Δ* cells, and fully rescues the end resection defect in a Sae2-dependent manner, but is unable to suppress the NHEJ, Tel1 signaling and replisome stabilization functions associated with loss of Xrs2 [189,195]. Purified MR is proficient for Sae2-stimulated endonucleolytic clipping at protein-blocked ends in vitro, consistent with the in vivo data showing that Xrs2 is dispensable for end resection [195]. By contrast, mammalian Nbs1 is important to regulate nuclease activity of the MR complex and is required for CtIP-stimulated endonucleolytic cleavage of substrates with 5′ adducts [176,177,194,196].

The Mre11 interaction domain is comprised of two regions: Interaction domain 1 wraps around the outside of the Mre11 phosphodiesterase domains in a highly extended conformation. Interaction domain 2 includes a highly conserved NFKxFxK motif that binds across the Mre11 dimer interface (Figure 3B). Interestingly, two Xrs2/Nbs1 molecules bind to an Mre11 dimer through interaction domain 1 while only one of the two Xrs2/Nbs1 molecules additionally binds to Mre11 via interaction domain 2. This second interaction is mediated by the eukaryotic-specific large loop insertion, referred to as the latching loop, within the phosphoesterase domain of Mre11 that is absent from bacteria and archaeal proteins [171,197]. It is suggested that the Xrs2/Nbs1 binding to the latching loop stabilizes the Mre11 dimeric form [171]. Indeed, expression of just a 108 amino acid fragment of murine Nbs1, encompassing the Mre11 interaction domain, is sufficient to sustain cell viability and improve dimer stability [194]. Interestingly, the 108 amino acid Nbs1 fragment is sufficient to restore CtIP-stimulated endonucleolytic cleavage by the MR complex in vitro, suggesting that the main function of Nbs1 in end resection is to stabilize the Mre11 dimer [194].

In addition to the Mre11 interaction domain, the C-terminal region of Xrs2/Nbs1 contains a conserved FXF/Y motif that mediates interaction with Tel1/ATM (Figure 3A) [15,16,17,18]. Deletion of the Tel1 interaction domain (TID) at the C terminus of Xrs2 (*xrs2-11*) results in a phenotype similar to the *tel1Δ* mutant, including defective Tel1-dependent DNA damage signaling and short telomeres [15]. However, ATM signaling is still detected in murine cells lacking the conserved C-terminal region of Nbs1 suggesting that there are additional contacts between ATM and MRN [194,198]. Fusion of the fission yeast Nbs1 TID to Mre11 or budding yeast Xrs2 TID to Mre11-NLS is able to restore telomere length and Tel1 signaling to *nbs1Δ* or *xrs2Δ* cells, respectively, indicating that the main role of Xrs2/Nbs1 in damage signaling is recruitment of Tel1/ATM to the MR complex [21,189]. In addition to DNA damage signaling, Tel1 plays a structural role to stabilize MRX-DNA association [199]. While this function of Tel1 does not normally contribute to the DNA repair activity of the MRX complex, hypomorphic mutation of MRX components that destabilize DNA binding synergize with *tel1Δ*, resulting in sensitivity to clastogens, loss of end tethering, replisome destabilization, and an increased rate of GCRs [188,189].

The N-terminal region of Xrs2/Nbs1 contains a forkhead-associated (FHA) domain, which binds to phosphorylated Sae2 /CtIP/Ctp1 and Lif1/Xrcc4 [56,57,58,200,201,202,203]. The FHA domain is directly fused to a tandem BRCA1 C terminus (BRCT) domain [201,204]. In mammals, checkpoint adaptor MDC1 interacts simultaneously with FHA and BRCT domains to engage with γH2AX and amplify the DNA damage checkpoint [2,205]. Although the Xrs2 FHA domain is known to interact with phosphorylated Sae2, this interaction appears to be unimportant for resection initiation in *S. cerevisiae* [195]. In contrast, interaction between Ctp1 (the functional Sae2/CtIP ortholog in fission yeast) and Nbs1 is critical for stable association of Ctp1 at DSBs, and an *nbs1* mutant defective for this interaction displays a phenotype similar to the *ctp1Δ* mutant [203].

## 4. Sae2/Ctp1/CtIP and the MRX Complex

*SAE2* was originally identified by its requirement to process Spo11-bound DSBs in meiosis, and subsequent studies showed that the *sae2Δ* mutant is also defective for hairpin resolution, a phenotype shared by *mre11-nd* and *rad50S* mutants [79,206,207]. In vitro, Sae2 and CtIP have been shown to modulate the nucleolytic activity of the MRX/N complex [25,26]. Sae2/CtIP requires CDK-dependent phosphorylation in order to stimulate the Mre11 endonuclease [25,26,77,184,208]. In addition to regulating the Mre11 endonuclease, a recent in vitro study showed that CDK-phosphorylated Sae2 also stimulates the exonucleolytic activity of MRX at a DNA nick [33]. At a mechanistic level, it is still unknown how Sae2/CtIP stimulates the endo- and exonuclease activities of Mre11, but likely involves interaction with Rad50 [184].

Independent of its stimulatory effect on Mre11 endonuclease, recombinant Sae2 has been reported to harbor an intrinsic ssDNA endonuclease that processes hairpin DNA substrates in vitro [209]. CtIP also has been shown to possess a 5′ flap endonuclease activity on branched DNA structures in vitro [210,211]. However, nuclease-free recombinant Sae2 and CtIP have also been reported [25,26,33,90], and no nuclease activity is associated with Ctp1 [212]. Further studies are required to verify the intrinsic nuclease activity of Sae2/Ctp1/CtIP and the contexts in which it acts.

Sae2 and its orthologs are largely unstructured proteins with few defined folded domains typical of nucleases and other DNA metabolizing enzymes [213]. X-ray crystal structures of Ctp1 and CtIP have revealed a conserved N-terminal oligomerization fold that assembles into a stable homotetramer (dimer of dimers), and this architecture is essential for end resection and repair by HR [212,214]. The oligomerized form of Sae2, Ctp1, and CtIP is structurally capable of binding and bridging DNA [212,214]. Consistently, purified Ctp1 shows effective DNA linking activity in vitro [212]. The *sae2Δ* mutant exhibits a defect in end tethering, but it is less severe than observed for *mre11Δ* or *rad50Δ* mutants, and the *sae2Δ mre11Δ* mutant behaves the same as *mre11Δ* indicating that Sae2 does not function independently of MRX to tether ends [186,215,216]. The precise roles of the two DNA-tethering activities within MRX/N-Sae2/Ctp1/CtIP complex in coordinating DNA molecules are currently unknown. The other conserved region of Sae2/Ctp1/CtIP lies at the C-terminus of the protein. This region contains an RNR/RHR motif important for DNA binding, as well as the critical CDK phosphorylation site in Sae2 and CtIP for activation of Mre11 endonuclease [25,26,212].

Intrinsically disordered proteins are often subjected to post-translational modifications that serve as reversible switches to control protein–protein interactions. Sae2 is phosphorylated on multiple sites during the cell cycle and in response to DNA damage [77,184,217,218]. As noted above, CDK-dependent phosphorylation of Sae2 activates Mre11 endonuclease in vitro and is required for meiotic recombination. Mec1 and/or Tel1-dependent phosphorylation of Sae2 is not essential for stimulating Mre11 endonuclease activity in vitro or in vivo, but contributes to activity by promoting oligomerization [184]. Phosphorylation of Sae2 by Mec1 and/or Tel1 promotes interaction with several FHA-containing proteins, including Xrs2, Rad53, and Dun1 [200]. However, the physiological significance of these interactions is not fully understood. In addition to phosphorylation, Sae2 is sumoylated in response to DNA damage and this modification is required for optimal Sae2 function [219]. Sae2 is also acetylated, and inhibition of histone deacetylases triggers its degradation resulting in altered MRX turnover and reduced end resection [220].

In vivo, the absence of Sae2 does not significantly impair resection at endonuclease-induced DSBs because the presence of MRX is sufficient for the direct recruitment of Exo1 and Dna2-Sgs1, which can initiate resection at ‘clean’ ends [28,81,215]. Interestingly, *sae2Δ* cells are less resistant to DNA damaging agents and exhibit a more pronounced resection defect than *mre11-nd* cells [28,81], and *sae2Δ* is lethal when combined with an *sgs1Δ* mutation whereas the *mre11-nd sgs1Δ* double mutant is viable [51,221]. Sae2 is required for the proper turnover of MRX at DSBs: Mre11 foci persist longer at damaged sites and the enrichment of Mre11 in the vicinity of a DSB is significantly higher in *sae2Δ* cells [9,43,78]. *mre11-nd* cells also exhibit persistent and hyper-enrichment of Mre11 at DSB ends, suggesting that the nuclease activity of Mre11 is required for timely eviction of the MRX complex [9,78]. However, *sae2Δ* cells exhibit higher Rad9 binding in the vicinity of DSBs than observed in the absence of Mre11 nuclease, and Rad53 is hyper-activated resulting in reduced resection by Dna2-Sgs1 and Exo1 [221,222,223]. Mutations that decrease Rad9 binding suppress *sae2Δ* DNA damage sensitivity by activating Sgs1-Dna2 dependent resection [199,223]. Several point mutations within *MRE11* have also been identified as suppressors of *sae2Δ* DNA damage sensitivity. One class of *mre11* suppressor mutations encode proteins with reduced DNA binding affinity and suppress Mre11 hyper-accumulation at DNA ends in the *sae2Δ* background resulting in reduced checkpoint signaling [224,225]. Another *mre11* mutation was recently characterized that accelerates Exo1-dependent resection in wild type and *sae2Δ* cells [226]. These studies indicate that Sae2 has a function in end resection independent of Mre11 nuclease activation that counteracts the negative effect of the DNA damage checkpoint on resection by Exo1 and Dna2-Sgs1.

In contrast to budding yeast, depletion of Ctp1 or CtIP severely impairs resection and HR [70,83,227]. Furthermore, deletion of CtIP in mouse leads to early embryonic lethality, indicating that CtIP is required for cell proliferation [228]. Mutation of just the C-terminal CDK site of CtIP confers embryonic lethality in mice, consistent with the requirement for Mre11 endonuclease activity for mammalian cell proliferation [20,229]. While Sae2 is thought to function primarily through the MRX complex, a recent study identified an Mre11 independent function of CtIP in protecting replication forks from Dna2-mediated degradation [230]. This study raises the possibility that CtIP, which is considerably larger than Sae2, has evolved additional functions in chromosome metabolism.

## 5. Concluding Remarks

Structural, biochemical and genetic studies over the last two decades have provided significant insight into the multiple functions of the MRX/N complex in maintaining genome integrity. However, many questions remain. For example, how Sae2/CtIP modulates the Mre11 endonuclease to initiate clipping of 5′-terminated strands at DSBs is not fully understood. The precise architecture of the MRX/N complex in tethering DNA molecules and the how it coordinates with the end bridging activity of Sae2/Ctp1/CtIP also remains to be elucidated. Although we have structural details of the core catalytic domains of the complex, we do not have a clear picture of the entire complex and how ATP conformational changes to the Rad50 subunit influence the other components and activation of Tel1/ATM. Finally, the function of MRX/N during replication is poorly understood and this is likely to be an active area of investigation in the future.

## Figures and Tables

**Figure 1 genes-09-00589-f001:**
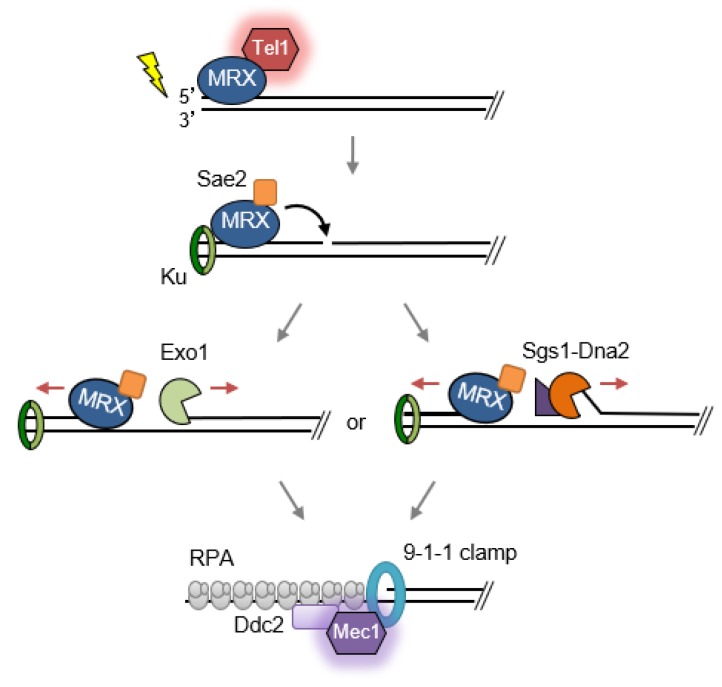
Damage recognition, end resection, and checkpoint activation. The Mre11-Rad50-Xrs2 (MRX) complex detects double-strand breaks (DSBs) and binds to the break ends (only one end is shown). Xrs2 recruits Tel1 and checkpoint signaling is activated. Resection follows a two-step, bidirectional mechanism. MRX, together with its cofactor Sae2, initiates resection by endonucleolytic cleavage of the 5′-terminated strand, generating an entry site for long-range resection machineries, Exo1 and Sgs1-Dna2, to proceed in the 5′ to 3′ direction. Meanwhile, the MRX complex proceeds back towards the double-stranded DNA (dsDNA) end using its 3′ to 5′ exonuclease activity. Single-stranded DNA (ssDNA) generated by resection is coated by replication protein A (RPA), which recruits Ddc2 and the Mec1 checkpoint kinase.

**Figure 2 genes-09-00589-f002:**
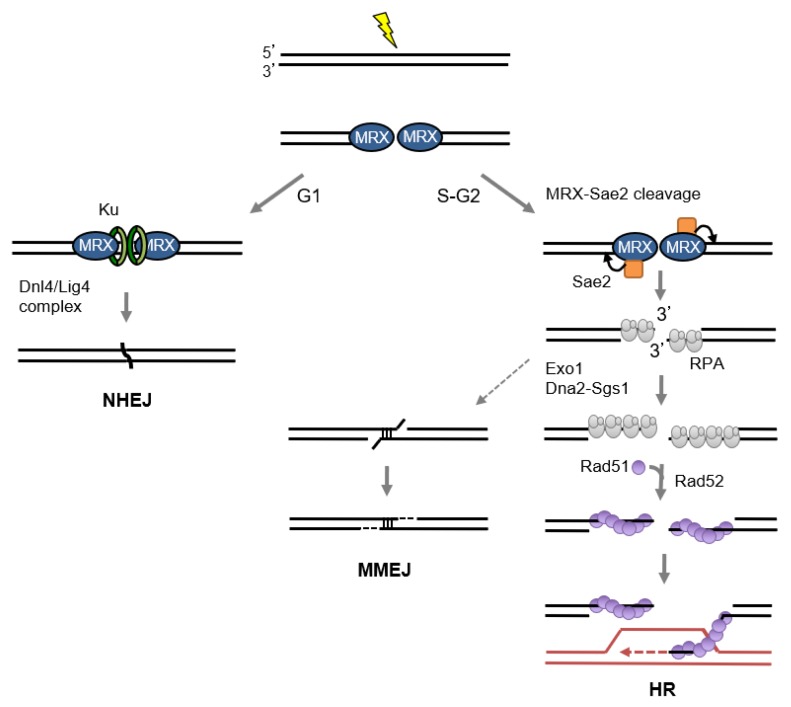
Overview of the DSB repair mechanisms. DSBs are repair by one of two major pathways: non-homologous end joining (NHEJ) or homologous recombination (HR). Classical NHEJ (C-NHEJ) directly re-ligates the two ends together while HR utilizes homologous template and is active in the S and G2 phases of the cell cycle when a sister chromatid is available. RPA initially binds to the 3′ ssDNA overhangs produced by end resection and is then replaced by Rad51 in a reaction requiring the Rad52 or BRCA2 mediator protein. The Rad51-ssDNA complex promotes the homology search and strand invasion, pairing the invading 3′ end with one strand of the donor duplex to template DNA synthesis. Resected intermediates can also be channeled to the MH-mediated end joining (MMEJ) pathway.

**Figure 3 genes-09-00589-f003:**
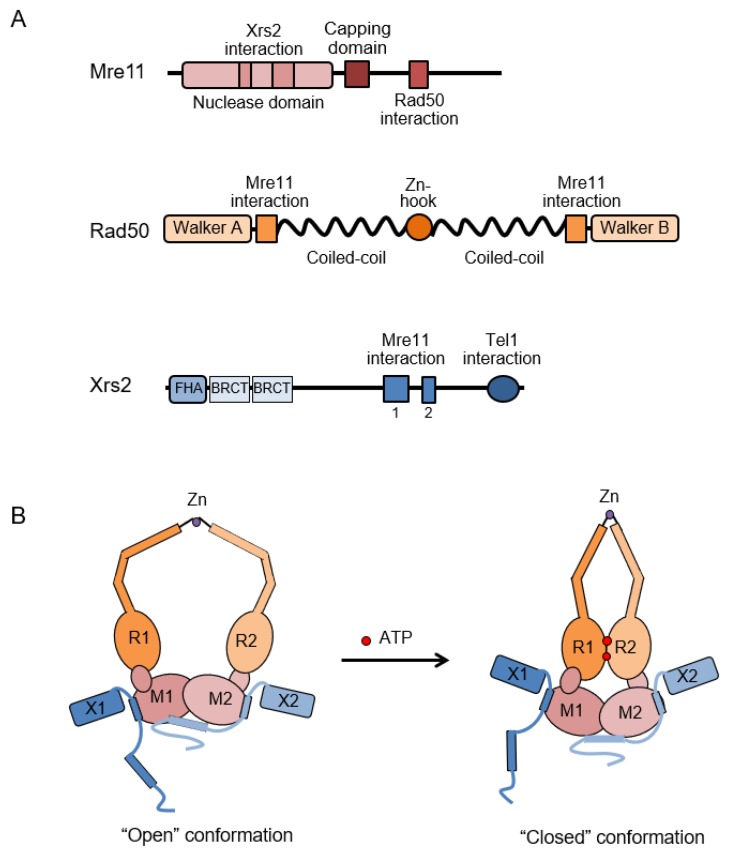
Domains and architecture of the MRX complex. (**A**) Mre11 consists a conserved phosphodiesterase nuclease domain and a capping domain at the N-terminus. The hydrophobic interaction domain for Rad50 resides towards the C-terminal region. Rad50 consists a bipartite ABC-ATPase domain at the N and C termini separated by two long coiled-coil domains and a zinc hook CxxC motif. Xrs2 harbors forkhead-associated (FHA) and BRCA1 C terminus (BRCT) domains at the N-terminus, and Mre11 and Tel1 interacting domains at the C-terminus. (**B**) Mre11, Rad50, and Xrs2 form a 2:2:2 heterohexameric complex, which undergoes a dramatic conformation change upon ATP binding. ATP binding by Rad50 induces the ‘closed’ form limiting access of the Mre11 nuclease active site to DNA. ATP hydrolysis opens the complex to allow Mre11 to initiate end processing.

**Figure 4 genes-09-00589-f004:**
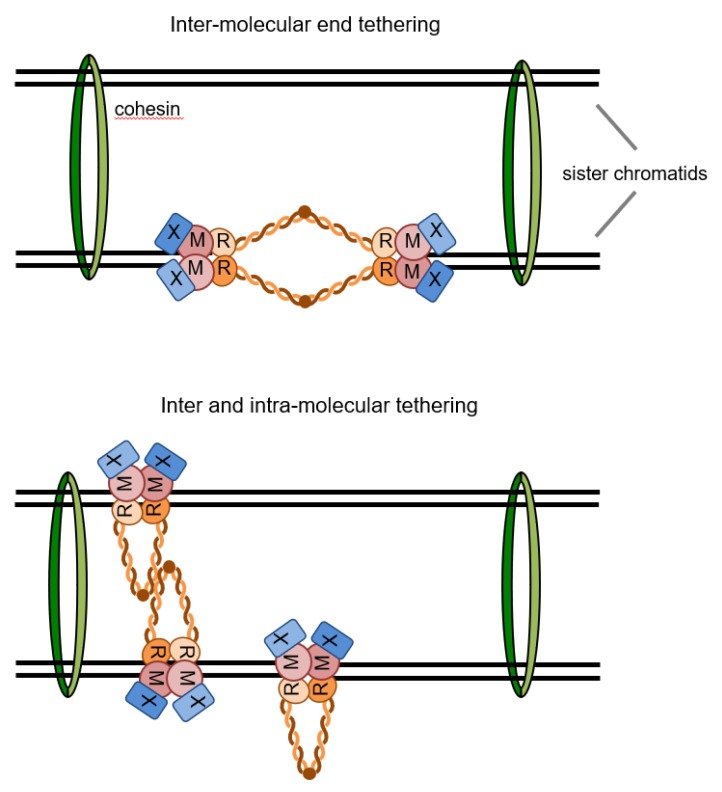
Potential end tethering configurations of the MRX complex. Assemblies of MRX at both DSB ends could bridge ends through intermolecular hook-mediated dimerization. Alternatively, intra-molecular dimerization of Rad50 could mediate bridging of DNA ends and oligomerization of the coiled-coils could facilitate sister chromatid interactions. In both cases, proximity to the sister chromatid is maintained through cohesin enrichment in the vicinity of DSBs.

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
