# Peer review of "Role of the Mre11 Complex in Preserving Genome Integrity"

_genes, 2018, doi:10.3390/genes9120589_

Round 1

Reviewer 1 Report

Manuscript Number:  Genes-397077  “ Role of the Mre11 complex in preserving genome integrity" by Julyun Oh and Lorraine S. Symington.

The review by Julyun Oh and Lorraine S. Symington analyzes the current status of literature on the role of the Mre11-Rad50-Xrs2/Nbs1 complex in the maintenance of genome stability in eukaryotic cells.  This is a thorough and accurate review of in vivo and in vitro activities of the complex. The authors methodically describe what is known about activities of the Mre11 complex in the checkpoint response cascade, NHEJ, DSB resection, mitotic and meiotic recombination, GCRs, telomere maintenance, and replisome stability.  Special attention the authors paid for the regulation mechanisms of the complex mediated by Sae2/Ctp1/CtIP interacting protein and CDK and Mec1 and Tel1 kinases. Recent discoveries on the structural analysis and in vitro properties if the Mre11 complex are also nicely included. In general, the reader gets a complete and up to date picture of the Mre11-Rad50-Xrs2/Nbs1 activities, structural features, and regulations.  There are several minor comments and suggestions outlined below:

1.    Line 40. I would include a specific reference for the NBS-like disorder resulting from mutations in RAD50:

Am J Hum Genet. 2009 May;84(5):605-16. doi: 10.1016/j.ajhg.2009.04.010. Epub 2009 Apr 30.

Human RAD50 deficiency in a Nijmegen breakage syndrome-like disorder. Waltes R1, Kalb R, Gatei M, Kijas AW, Stumm M, Sobeck A, Wieland B, Varon R, Lerenthal Y, Lavin MF, Schindler D, Dörk T.

2.     Line 103.  In light of studies from J. Strathern and J. Haber labs on the mutagenic features of homologous recombination, I would not define HR as “accurate repair.”

3.    Line 198. Studies from GR Smith on palindrome stability in S. pombe and the role of the MRX complex should be included

4.    Following the flow of the text, I would move up the last paragraph in the “Mre11 and Rad50” section (lines 332-344). This description outlined in Figure 3 comes after Figure 4.

5.    The role of RPA in stimulating harping opening by the MRX-Sae2 should be mentioned:

Genes Dev. 2017 Dec 1;31(23-24):2331-2336. doi: 10.1101/gad.307900.117. Epub 2018 Jan 10.Plasticity of the Mre11-Rad50-Xrs2-Sae2 nuclease ensemble in the processing of DNA-bound obstacles.Wang W1, Daley JM1, Kwon Y1, Krasner DS1, Sung P.

6.    Line 420. “phosphorylation” word is missing

Author Response

All the suggestions made by Reviewer 1 have been incorporated into the revised manuscript:

1.     The suggested reference has been added

2.     Mostly was added before accurate.

3.     The reference to G. Smith’s work on palindrome stability in S. pombe has bee added.

4.     The paragraph was reorganized as suggested.

5.     The role of RPA in stimulating hairpin opening by MRX-Sae2 was added to the section on hairpin resolution.

6.     Phosphorylated was added to the sentence.

Reviewer 2 Report

Review is complete and well structured. Figures help understand complex processes. I recommend publication in the present form. 

Author Response

Reviewer 2 had no comments to be addressed